# Laparoscopic vs. Open Gastrectomy for Locally Advanced Gastric Cancer: A Propensity Score-Matched Retrospective Case-Control Study



Stefano Caruso [1,*], Rosina Giudicissi [2], Martina Mariatti [1], Stefano Cantafio [2], Gian Matteo Paroli [1] and Marco Scatizzi [1]

[1] Department of General Surgery and Surgical Specialties, Unit of General Surgery, Santa Maria Annunziata Hospital, Central Tuscany Local Health Company, Via dell'Antella 58, Bagno a Ripoli, 50012 Florence, Italy; martina.mariatti@uslcentro.toscana.it (M.M.); gianmatteo.paroli@uslcentro.toscana.it (G.M.P.); marco.scatizzi@uslcentro.toscana.it (M.S.)

[2] Department of General and Oncologic Surgery, Unit of General Surgery, Santo Stefano Hospital, Central Tuscany Local Health Company, 59100 Prato, Italy; rosina.giudicissi@uslcentro.toscana.it (R.G.); stefano.cantafio@uslcentro.toscana.it (S.C.)

* Correspondence: caruso.stefano1976@gmail.com; Tel.: +39-55-9508373 or +39-349-8312397

**Abstract:** Introduction: Minimally invasive surgery has been increasingly used in the treatment of gastric cancer. While laparoscopic gastrectomy has become standard therapy for early-stage gastric cancer, especially in Asian countries, the use of minimally invasive techniques has not attained the same widespread acceptance for the treatment of more advanced tumours, principally due to existing concerns about its feasibility and oncological adequacy. We aimed to examine the safety and oncological effectiveness of laparoscopic technique with radical intent for the treatment of patients with locally advanced gastric cancer by comparing short-term surgical and oncologic outcomes of laparoscopic versus open gastrectomy with D2 lymphadenectomy at two Western regional institutions. Methods: The trial was designed as a retrospective comparative matched case-control study for postoperative pathological diagnoses of locally advanced gastric carcinoma. Between January 2015 and September 2021, 120 consecutive patients who underwent curative-intent laparoscopic gastrectomy with D2 lymph node dissection were retrospectively recruited and compared with 120 patients who received open gastrectomy. In order to obtain a comparison that was as homogeneous as possible, the equal control group of pairing (1:1) patients submitted to open gastrectomy who matched those of the laparoscopic group was statistically generated by using a propensity matched score method. The following potential confounder factors were aligned: age, gender, Body Mass Index (BMI), comorbidity, ASA, adjuvant therapy, tumour location, type of gastrectomy, and pT stage. Patient demographics, operative findings, pathologic characteristics, and short-term outcomes were analyzed. Results: In the case-control study, the two groups were clearly comparable with respect to matched variables, as was expected given the intentional primary selective criteria. No statistically significant differences were revealed in overall complications (16.7% vs. 20.8%, $p = 0.489$), rate of reoperation (3.3% vs. 2.5%, $p = 0.714$), and mortality (4.2% vs. 3.3%, $p = 0.987$) within 30 days. Pulmonary infection and wound complications were observed more frequently in the OG group (0.8% vs. 4.2%, $p < 0.01$, for each of these two categories). Anastomotic and duodenal stump leakage occurred in 5.8% of the patients after laparoscopic gastrectomy and in 3.3% after open procedure ($p = 0.072$). The laparoscopic approach was associated with a significantly longer operative time (212 vs. 192 min, $p < 0.05$) but shorter postoperative length of stay (9.1 vs. 11.6 days, $p < 0.001$). The mean number of resected lymph nodes after D2 dissection (31.4 vs. 33.3, $p = 0.134$) and clearance of surgical margins (97.5% vs. 95.8%, $p = 0.432$) were equivalent between the groups. Conclusion: Laparoscopic gastrectomy with D2 nodal dissection appears to be safe and feasible in terms of perioperative morbidity for locally advanced gastric cancer, with comparable oncological equivalency with respect to traditional open surgery.





**Keywords:** gastric cancer; gastric resection; minimally invasive surgery; laparoscopic gastrectomy; open gastrectomy

## 1. Introduction

Total and distal gastrectomy with D2 lymph node dissection is currently the recommended surgical procedure for resectable gastric cancer patients, apart from most early gastric cancer cases, in which endoscopic treatment or a limited lymphadenectomy (D1 or D1+) has been widely accepted [1–3].

Enhanced postoperative recovery of minimally invasive surgery is related to reduction of surgical trauma. Over the years, advances in minimally invasive surgery have caused a paradigm shift towards laparoscopic procedures.

Several reports have provided level III evidence that laparoscopic-assisted distal gastrectomy is technically safe and that it yields better short-term outcomes than conventional open gastrectomy (OG) for early-stage gastric cancer [4–16]. Laparoscopic-assisted gastrectomy for distal early-stage gastric cancer has progressively spread worldwide, especially in Eastern countries, and presently it is the standard therapy in Asian countries, such as Japan and Korea [17,18]. On the other hand, a safer laparoscopic gastrectomy (LG) with D2 resection technique for the treatment of advanced gastric cancer (AGC) did not meet the same acceptance. The widespread diffusion of laparoscopic surgery to manage AGC is limited mainly by the technical difficulties posed by total gastrectomy and D2 lymphadenectomy.

Gradually, with the passing of time, together with the improvement of laparoscopic technology, an increasing number of surgeons have demonstrated their skill in performing total gastrectomy and adequate laparoscopic D2 lymphadenectomy [19–21] in advanced gastric tumours, and data on long-term survival are increasingly being published [22–28].

However, available articles mainly originate from Eastern countries, and large-cohort studies are still scarce worldwide to date. This has given rise to concern, especially in Western countries, regarding the oncological adequacy and long-term outcomes of laparoscopic surgery for AGC.

Safety and oncological adequacy of a new technique have to be ensured before the method can be widely recommended. The commitment is to know whether LG can effectively reproduce the same procedure as performed in open surgery and obtain the same oncological results. In the case of advanced gastric tumour, this basically concerns the adequacy of lymphadenectomy, the suitability of gastric resection (with free margin), and the ability to complete the reconstruction (especially after total gastrectomy). In order to help address the shortage of evidence with regards these concerns, we report our experience resulting from a dual Western centre series of curable AGC patients, retrospectively compared to the traditional open technique using a statistically generated propensity score-matched case-control method. The comparison focused on the analysis of short-term surgical outcomes and oncological adequacy of LG vs. OG with D2 lymphadenectomy.

## 2. Methods

### 2.1. Study Design

The present population-based cohort study was a propensity score-matched case-control study comparing two treatment arms (laparoscopic vs. open gastrectomy) for gastric cancer. The data derived from an electronic prospectively maintained database of patients submitted to surgery in two regional Italian institutes renowned for their high proficiency in oncological procedure: the Santa Maria Annunziata Hospital of Florence and the Santo Stefano Hospital of Prato. Both aforementioned hospitals work for the 'Central Tuscany Local Health Company'. Data collection was performed in each hospital with an identical preformed module and then unified in a common gastric cancer database. The database included information on demographics, clinical, surgical, short-term outcomes, and long-term follow-up results for all consecutive patients who underwent curative

gastrectomy for gastric cancer at the participating institutions regardless of the type of procedure (laparoscopic or open).

We focused on only short-term outcomes as a preliminary investigation that can hopefully be the basis for a future long-term analysis. The purpose was to assess the technical feasibility and the oncologic non-inferiority of laparoscopic gastrectomy with D2 dissection compared to conventional open D2 gastrectomy for advanced-stage gastric cancer.

The whole study group consisted of patients treated at our institutions with intent-to-cure surgery during the period from January 2015 to September 2021. The period previous to January 2015 was not considered adequate due to the failure of achieving an adequate learning curve. Our institutes exceed the number of 21 procedures per year performed for gastric cancer patients, which is associated with higher overall and disease-free survival with respect to lower-volume centres. This emphasizes the value of performing gastric cancer surgeries in high-volume centres, similar to the Western world [29]. At our regional surgical centre, the first laparoscopic distal gastrectomy was performed in January 2006 (by M.S.). Initially, operations were performed exclusively by a single surgeon (M.S.) with extensive experience in gastric cancer surgery and general laparoscopic surgery. Successively, LG has been progressively and increasingly applied over time at both our institutions, and thus other professionals on our team have gained skills in this regard. Currently, the laparoscopic procedure is the first-line modality at our institutions in the absence of contra-indications. However, the surgeons experienced in laparoscopic procedures have not completed the learning curve at the same time, and not all surgeons of the team have achieved adequate performative skills in LG. Indeed, a certain proportion of cases, even if preponderant in the earlier period, were performed with the traditional open technique.

Therefore, the decision to operate using a laparoscopic or open approach for patients diagnosed preoperatively with gastric cancer was not randomized at our institutions. Mainly it depended on the expertise level achieved by the surgeons at the time of surgery, together with patient characteristics on a case-by-case basis.

All included procedures were performed or supervised at the time of their execution by the surgeon in charge at the highest level of experience (usually M.S.), highly trained in laparoscopic oncology surgery, together with a team adequately proficient in either the open or laparoscopic technique. As suggested by the range reported in the literature that can be considered satisfactory for the achievement of an adequate learning curve, the attainment of a case volume of 40 LGs for the main surgeon conducting the operation in order to be included in this study was considered satisfactory. Thus, balanced by the chosen study period, which for this reason starts from 2015, the comparison between the LG and OG is methodologically guarded from potential learning curve bias.

### 2.2. Patient Population

All patients underwent diagnostic and preoperative staging work-up according to a standard protocol, which included upper digestive endoscopy with gastric biopsy and computed tomography of the abdomen and chest. The evaluation of the preoperative diagnostic process and the successive management decisions were discussed at regular weekly multidisciplinary team meetings (oncologist, surgeon, radiotherapist, gastroenterologist, pathologist, and radiologist).

All items regarding patients' characteristics (age, gender, body mass index [BMI], co-morbidities, history of abdominal surgery, American Society of Anesthesiology risk class [ASA]), process of care (comprising preoperative diagnostic work-up evaluation of multidisciplinary team meetings and eventual neo-adjuvant chemo-radiotherapy), operative characteristics (type of gastric resection, digestive reconstruction, associated procedures, operative time), pathological tumour features (location of the tumour, Lauren's histotype, histological differentiation grade, number of lymph nodes harvested and nodal status, disease stage, margin involvement), and hospital course (intra- and post-operative morbidity and mortality, reoperation, and length of stay) were revised for the present study.

Written informed consent was obtained from all of the patients prior to their operations in accordance with the ethical guidelines of our company (Clinical Ethics Committee, Central Tuscany Local Health Company, Florence, Italy).

### 2.3. Enrollment Process and Definition of Comparative Groups

We initially screened from the electronic database all medical records of patients who had undergone laparoscopic or open gastrectomy for gastric cancer during the period from January 2015 to September 2021.

Within this study period, a total of 420 gastric cancer patients were submitted to surgery at our two centres and were initially retrieved. Then, the enrollment process continued in order to identify those patients undergoing curative intent gastrectomy with D2 lymph node dissection for locally advanced gastric carcinoma.

Specifically, cases considered suitable for the comparison followed these inclusion criteria: (1) histologically proven adenocarcinoma of the stomach through endoscopic biopsy, (2) pathologically definitive staging of II–III according to the Union for International Cancer Control (UICC) TNM Classification of Malignant Tumours, 8th Edition [30] (hence T4b tumours that required organ resection were not excluded), (3) no evidence of distant metastasis (by means of preoperative and intraoperative staging), (4) curative intent surgery (R0-1, defined respectively as no tumour cells or microscopic residual at postoperative pathology). Exclusion criteria were: (1) pathology negative for adenocarcinoma, (2) early-stage gastric cancer (pStage I), (3) evidence of distant metastasis (pStage IV, including para-aortic lymph node involvement), (4) R2—macroscopic residual disease at the end of operation (palliative intent surgery), (5) previous surgery for gastric cancer, (6) ASA—American Society of Anesthesiologists score > 3. Patients who had undergone neo-adjuvant chemotherapy were not excluded. The vast majority of cases received regimens similar to the MAGIC-trial (Epirubicin, Cisplatin, and Capecitabin) [31].

Ultimately, after application of inclusion and exclusion criteria, the data of 306 patients submitted to a potentially curative (R0/R1) gastrectomy with D2 lymph node dissection for primary gastric adenocarcinoma were identified. We had to exclude 114 patients from the cohort: eight patients did not have adenocarcinoma, eight had a history of surgery for gastric cancer, 44 patients presented with stage I cancer, 19 patients presented with stage IV cancer, 11 patients had an ASA score higher than 3, 14 patients had undergone gastrectomy with residual macroscopic tumour (R2) at the end of (palliative) surgery, and 10 patients had missing data for estimation of the propensity score. A flowchart of patient enrollment is shown in Figure 1. The remaining patients were recruited for the study, and cases who underwent laparoscopic and open gastrectomy were divided. Those LG cases meeting the required criteria, constituting the LG group (*n* = 120), were subsequently subjected to a propensity score matching for comparison to a 1:1 pairing of matched patients who underwent the same procedure in an open fashion during the same period. The patient pairing method and the alignment of groups for potential confounding factors are described in detail in the statistical analysis section. In this way, a control group (OG group) consisting of an equal number (*n* = 120) of balanced patients was generated.

### 2.4. Outcome Measures

Histopathological, surgical, and short-term oncologic outcomes were compared between the LG and OG groups. Primary endpoints were short-term outcomes, including morbidity, postoperative complications, postoperative hospital stay, and mortality. Postoperative complications and mortality were defined as complications or death within 30 days of surgery or during hospitalization. Intra-operative and post-operative complications were graded according to the Clavien–Dindo classification [32,33].

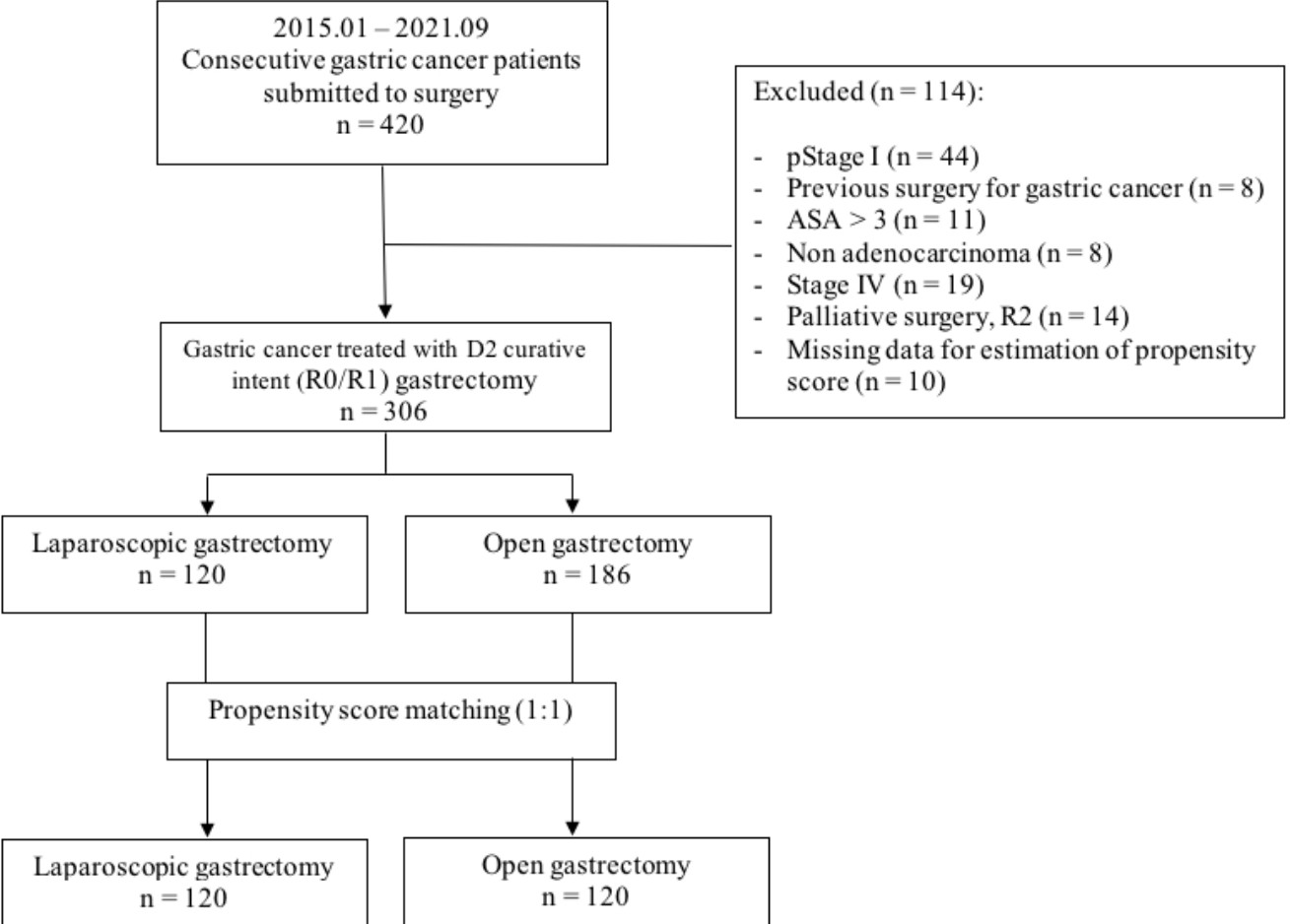

**Figure 1.** Flowchart of patient enrollment.

Most patients in both groups began mobilization and if possible, walking the first day after surgery, in accordance to an ERAS (Enhanced Recovery After Surgery) protocol, which has been formally approved for years at our institutions for colorectal surgery and successively extended to gastric surgery. This protocol had already been introduced in one of our institutes (Prato) before 2015, the time when data extraction for this study began, while in the other (Florence), it was introduced in 2018.

Anastomotic leakage was defined as a clinical or radiological diagnosed leak, including duodenal stump leaks. Pulmonary complications included pneumonia (lung infiltrate on X-ray), pleural effusion, and acute respiratory distress. Wound complications comprised infection or abscess and fascia dehiscence. Abdominal abscess was defined as a purulent discharge with positive cultures obtained from abdominal drains placed during surgery or fluid collection requiring drainage.

Reoperation was defined as every surgical procedure following primary surgery during hospitalization or within 30 days after primary intervention.

The resected specimens, which included lymph node yield, and the radicality of the resection (R0-1) were reviewed by pathologists in accordance with the American Joint Committee on Cancer TNM staging system [30]. For the patients operated on between 2015 and 2017, the staging was converted and aligned with the latest TNM classification by AJCC (8th ed.). The tumours were also classified according to Lauren's histotype (i.e., intestinal, diffuse, or mixed) and for histologic differentiation grade (well differentiated; moderately differentiated; poorly differentiated; undifferentiated or signet-ring cell carcinoma).

*2.5. Surgical Procedure*

All patients underwent radical, subtotal or total, gastrectomy with D2 lymph node dissection, while total omentectomy was performed when serosal invasion was evident, according to guidelines of the Japanese Gastric Cancer Association [2] (5th ed. and previous). Pancreatectomy (distal) and splenectomy had not been performed routinely unless necessary (when there was direct tumour invasion or for iatrogenic cause).

The surgical procedure included distal gastrectomy and total gastrectomy, depending on tumour location and macroscopic characteristics. Subtotal distal gastrectomy was performed for tumors located in the middle or lower third of the stomach if sufficient proximal resection margin could be obtained. A minimum of a 3 to 5 cm proximal cancer-free margin can be achieved depending upon the depth of invasion and the growth pattern of the cancer (in accordance with Japanese gastric cancer treatment guidelines (5th ed. and previous) [2]. Total gastrectomy was performed for tumours in the proximal third of the stomach or those of the middle third in which a satisfactory proximal resection margin could not be obtained. For tumours of the cardias (Siwert I–II) invading the esophagus, frozen section examination of the resection line was carried out whenever a macroscopically adequate distance from the tumour was not certain to ensure an R0 resection. Siwert I tumors were considered equal to esophageal cancer and were therefore excluded from this series.

In the laparoscopic procedure, a carbon dioxide ($CO_2$) pneumoperitoneum of ~12 mm Hg was created by insufflation via the open supra-umbilical Hasson technique or Veress needle inserted at Palmer's point, with the patient placed in the supine split leg, reverse Trendelenburg position, with the operating surgeon between the patient's legs, a camera assistant and the assistant surgeon on the left side. After establishment of pneumoperitoneum and introduction of the camera port, the working ports and assistance ports were introduced under laparoscopic vision. The type and placement of trocars, number of ports, and location of the mini-laparotomy incision for extracting the resected specimen were discretionary, depending on patients' clinical characteristics (abdominal conformation, BMI, tumour size, tumour location, etc.) and according to the surgeons' preference. Commonly, four 12 mm laparoscopic trocars were inserted as displayed in Figure 2: one supra-umbilical trocar for the (30° forward-oblique) laparoscope, one trocar in the left mid clavicular line at the intersection with the transverse umbilical line and one at the contralateral site, and one trocar in the epigastrium at the right paramedian line 2 cm below the costal margin.

Open interventions were performed through a bilateral subcostal or median incision depending on the patient's anatomical configuration.

Independently from the laparoscopic or open technique, the same phases of surgery were carried out, with the first step consisting of the exploration of the abdominal cavity, aimed at excluding the presence of peritoneal seeding or any metastasis eventually not identified preoperatively.

Once the effective resectability of the neoplasm was confirmed, the essential phases of dissection were carried out using a laparoscopic energy device (usually ultrasonic shears—Ultracision–Harmonic[TM] Scalpel, Ethicon Endo-Surgery Inc., Cincinnati, OH, USA—or radiofrequency device—Ligasure[TM] Instrumets Covidien, Medtronic, Minneapolis, MN, USA—or hybrid technology, ultrasonic and bipolar—Thunderbeat[TM], Olympus Corporation, Tokyo, Japan.

The D2 lymphectomy included the removal of D1 lymph nodes (stations from n. 1 to 7) plus stations 8a, 9, 11p, 12a, and 14v for distal gastrectomy, with the addition of stations n. 10 and 11d for total gastrectomy, according to the Japanese gastric cancer treatment guidelines (5th ed. and previous) [2].

As already partially described elsewhere [34], the essential steps were as follows. After the gastrocolic ligament was divided along the border of the transverse colon, the dissection of the left greater omentum was performed in order to remove it 'en bloc' with the stomach, and lymph nodes along the left gastroepiploic vessels (n. 4sb) were dissected.

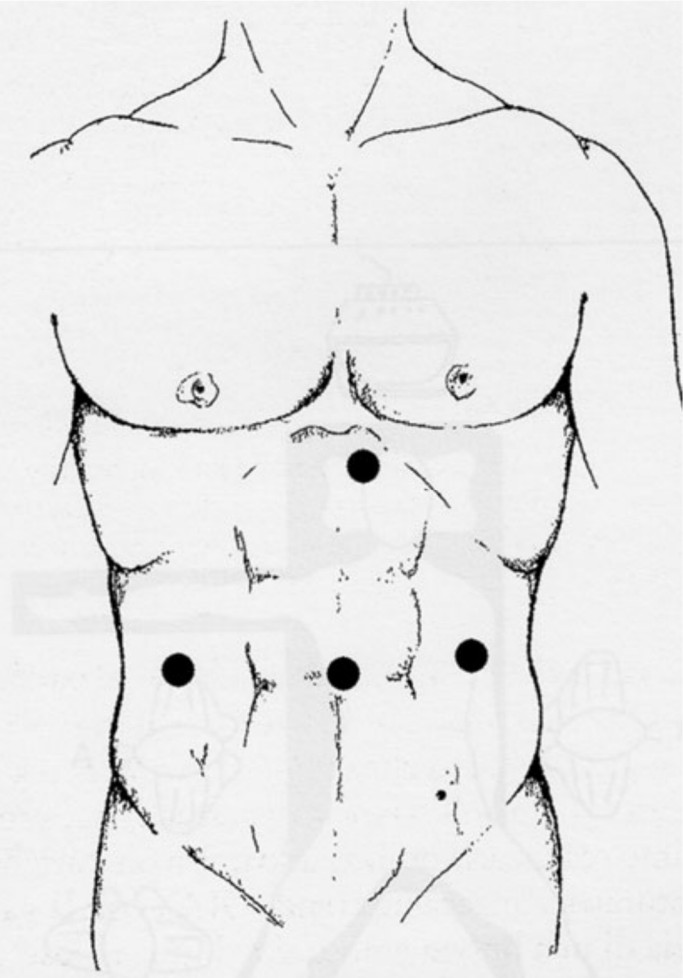

**Figure 2.** Trocar placement for laparoscopic gastrectomy.

The right omentum and lymph nodes along the right gastroepiploic vessels (n. 4d) were dissected, continuing the procedure to the right toward the pylorus, until the identification of the gastroduodenal artery between the medial side of the duodenal and pancreas head.

After exposure of Henle's trunk, the right gastroepiploic vessels were ligated at the inferior border of the pancreas, and the infra-pyloric lymph node (n. 6) and the nodes along the superior mesenteric vein (n. 14v) were dissected. Transection of the duodenum was performed with a linear stapler (45 mm or 60 mm endoscopic stapling device) 1–2 cm distally to the pylorus. The lesser omentum was divided and the dissection continued to the right along the right gastric artery, which was then exposed and divided at its origin with double clips, with the dissection of the suprapyloric lymph nodes (n. 5).

After the incision of the hepatogastric ligament, the dissection continued anterior to the hepatoduodenal ligament with the removal of lymph nodes of the proper hepatic artery (n. 12a) and then along the common hepatic artery (n. 8). The proximal splenic artery lymph nodes (n. 11p) were excised together with the surrounding fatty connective tissues. The lymph nodes along the distal splenic artery (n. 11d) and splenic hilum (n. 10) were removed in total gastrectomy or when macroscopically enlarged.

The dissection proceeded cephalad towards the gastroesophageal junction along the lesser curvature, during which a simultaneous D1 lymphadenectomy of the perigastric nodes was performed (n. 3, lesser curvature) until the esophagogastric junction was reached and the right cardiac nodes (n. 1) approached.

The stomach was lifted up towards the head to expose the gastropancreatic fold. The left gastric vein was prepared and separately divided at the upper border of the pancreatic

body, and then the root of the left gastric artery was identified, thus doubly clipped and divided at the origin, followed by dissection of n. 7 nodes and the nodes around the celiac artery (n. 9).

An identical operative strategy was performed by dissecting the gastrocolic ligament toward the spleen and along the left greater omentum. Short gastric vessels were divided depending on the entity of gastrectomy usually using harmonic or advanced bipolar scissors. In case of total gastrectomy, the left gastroepiploic artery, posterior gastric artery, and all short gastric vessels were divided with either energy device scissors or clips, and the dissection of lymph node continued towards 4sa, and the left-cardia nodes (n. 2) were dissected as well. Left cardiac nodes (group 2) were not necessarily dissected for cancer of the distal third of the stomach.

In distal gastrectomy, the stomach was transected using a 60 mm (endoscopic) linear stapling device, and then Roux-en Y or Billroth-II method was performed for reconstruction. Hand-sewing or using staplers for anastomosis was not limited. In total gastrectomy, Roux-en Y reconstruction was carried out. An end-to-side oesophagojejunal anastomosis was carried out with a circular stapler (ECS 25 mm, Ethicon Circula Stapler$^{TM}$, J&J Medical Device, Cincinnati, OH, USA). After the incision of the phrenoesophageal membrane, the oesophagus was transected at the planned plan, and a continuous purse-string suture was created in the muscle layer of the oesophagus completed followed by anvil placement. The surgical specimen of the oesophageal margin was recovered for later retrieval or immediate intraoperative pathologic exam if a macroscopically adequate distance from the tumour was not certain. The jejunal stump was closed with an endoscopic linear stapler. Jejunal and duodenal stump closure may be reinforced by continuous absorbable hand sutures (usually 3/0 absorbable self-locking V-Loc$^{TM}$—Covidien, Medtronic, Minneapolis, MN, USA). The Roux-en-Y limb, lead via a trans-mesocolic route, was reconstructed by a side-to-side jejunojejunostomy using a (endoscopic) 45 mm linear stapler. In laparoscopic procedure, anastomosis restoring the continuity was usually performed in an intracorporeal fashion, and the enterotomy was closed with a 3/0 absorbable suture.

The removal of the resected specimen (including the stomach, omentum, and lymph nodes) occurred via a mini-laparotomy (max. 5–6 cm, usually slightly enlarging the left pararectal or the periumbilical port-site incision), which must be muscle sparing, using a specimen retrieval Endocatch bag or a wound protector (Alexis$^{®}$ Wound retractor, Applied Medical, Rancho Santa Margarita, CA, USA). This incision can be used to perform the extracorporeal jejunojejunal anastomosis when this choice is made, and then the abdomen is reinsuffled to complete the reconstruction. At the end of the operation, a Blake peritoneal drain was inserted through trocar orifices and usually placed in Morrison's pouch. The nasogastric tube in distal gastrectomy is not placed, while in total gastrectomy, a naso-jejunal feeding tube is usually inserted.

*2.6. Statistical Analysis*

We performed a one-to-one matching analysis between laparoscopic and open gastrectomy groups. The LG group represented the study population cohort under the primary (safety and feasibility) analysis. The OG group constituted the statistically generated case-control group, selected among the entire group of curative-intent gastrectomies performed at our institutions during the same period as the laparoscopic group. To generate a comparable and equally distributed control group, which would allow proper evaluation of the effectiveness of laparoscopic procedure, a comparison with the traditional open technique was adjusted from potential confounding factors through a propensity score matching analysis.

The propensity score for each patient was calculated using a multivariable logistic regression model [35], given the following individual covariates that might be considered as potential baseline confounders [36]: age, gender, BMI, comorbidity, ASA, adjuvant therapy, tumour location, type of gastrectomy (total or subtotal), and pT stage. Covariates were selected from all baseline characteristics (parsimonious model, developed using bagging) [37].

Nearest-neighbour matching without replacement was performed to generate matched pairs of cases (ratio = 1:1) in which the average within-pair difference in propensity scores was minimized by setting a caliper of 0.20 width multiplied by the standard deviation of the logit of the propensity score [38], except for the categorical variables of the pT stage that had been exactly matched to achieve better balance [39]. Subjects that remained unmatched under this width were then discarded.

In the statistical analysis of pair-matched data, continuous outcomes were compared using paired *t*-tests or the Wilcoxon signed rank test as appropriate; differences in proportions were compared using the McNemar's test. The R program (R Foundation for Statistical Computing version 4.1.2 open-source software, http://www.R-project.org; 'MatchIt' and 'optmatch' packages) was used for the propensity score match.

In baseline comparison between the two groups, continuous variables were displayed by the mean ± standard deviation (SD), and categorical variables were expressed as percentage differences. The Mann–Whitney U nonparametric test was used for univariate analysis of continuous nonparametric variables, whereas Fisher's exact test, $\chi^2$ test or an unpaired Student's *t* test as appropriate was selected to examine categorical and dichotomous variables. Data were analyzed on an intention-to-treat basis, and those cases converted to laparotomy approach were analyzed in the laparoscopic group.

Logistic regressions for in-hospital or 30-day specific types of postoperative complications were performed. Adjusted odds ratios (relative risks, RR) along with the corresponding 95% confidence intervals (CI) were calculated for the differences in postoperative complication rates using the univariate Cox proportional hazard model. $p < 0.05$ was considered statistically significant. Statistical analysis was performed using the SPSS software package with the exact test option (22.0, SPSS, Chicago, IL, USA).

## 3. Results

### 3.1. Baseline Characteristics and Surgical Outcome

Ultimately, after selection criteria and propensity score application, 240 patients (120 for each group, LG and OG) who had undergone gastrectomy with curative intent for gastric cancer were recruited for the present analysis. Clinical characteristics and tumour location are summarized in Table 1. Globally, 152 were male, and 88 were female (male/female ratio 1.7:1). The mean age of the patients was 70.8 ± 10.1 years (range, 40–90 years). The mean body mass index (BMI) of the patients was 20.8 kg/m$^2$ range, 17.5–36.1 kg/m$^2$). Slightly more than one third (83/240; 34.6%) of the patients had comorbidities, the most common being hypertension. Approximately ~10% of patients in each the two groups had undergone previous abdominal surgery, with no significant difference between the two groups ($p = 0.481$). Surgical risks were classified as ASA 2 (39% of patients) and ASA 3 (46%) in most cases. Operative and pathologic findings are summarized in Table 2.

As was expected based on the study's design, both groups were well balanced for the variables (age, gender, mean BMI, comorbidity, ASA, adjuvant therapy, tumour location, type of gastrectomy, pT stage) that were considered for propensity score matching (Tables 1 and 2), and the pT stage was perfectly superimposable between the LG and OG groups ($p = 1.000$) (Table 1).

Only a minority of patients of both groups (4.2% vs. 5.0%) had undergone neoadjuvant chemotherapy, with no significant difference ($p = 0.650$).

A total of three (2.5%) LG procedures were converted to OG: two for technical difficulties due to severe peritoneal adhesions, and one for uncontrolled bleeding from the spleen.

In total, 164 patients (68.3%) underwent a subtotal distal gastrectomy, 86 in the LG group and 78 in the OG group, while total gastrectomy represented the procedure in the remaining cases ($n = 76$, 34 in LG and 42 in OG).

**Table 1.** Patients' characteristics.

| Characteristics | LG (*n* = 120) | OG (*n* = 120) | *p* Value |
|---|---|---|---|
| Age, Mean ± SD (range, year) | 70.5 ± 11.4 (40–90) | 70.9 ± 8.5 (44–89) | 1.000 |
| Sex, *n* (%) | | | 1.000 |
| Male | 77 (64.1%) | 75 (62.5%) | |
| Female | 43 (35.9%) | 45 (37.5%) | |
| BMI, mean ± SD (range, kg/m$^2$) | 20.7 ± 1.2 (17.5–35.4) | 21.0 ± 1.5 (18.1–36.1) | 0.560 |
| Previous abdominal surgery, *n* (%) | 13 (10.8%) | 15 (12.5%) | 0.481 |
| Co-morbidity, *n* pts (%) | 41 (34.1%) | 42 (35.0%) | 1.000 |
| Cardiovascular | 3 | 4 | |
| Hypertension | 20 | 17 | |
| Diabetes | 5 | 10 | |
| Pulmonary | 4 | 2 | |
| Hepatic | 2 | 2 | |
| Renal | 4 | 2 | |
| Other | 3 | 5 | |
| ASA score, *n* (%) | | | 1.000 |
| 1 | 18 (15.0%) | 18 (15.0%) | |
| 2 | 46 (38.3%) | 48 (40.0%) | |
| 3 | 56 (46.7%) | 54 (45.0%) | |
| Tumour location | | | 1.000 |
| upper | 17 (14.2%) | 19 (15.8%) | |
| mid | 46 (38.3%) | 48 (40.0%) | |
| lower | 57 (47.5%) | 53 (44.2%) | |
| Neoadjuvant Chemotherapy (%) | | | 0.650 |
| Yes | 5 (4.2%) | 6 (5.0%) | |
| No | 115 (95.8%) | 114 (95.5%) | |

LG: laparoscopic gastrectomy; OG: open gastrectomy; BMI: body mass index; ASA: American Society of Anesthesiologists. SD: standard deviation

**Table 2.** Operative findings and pathologic outcomes.

| Characteristics | LG (*n* = 120) | OG (*n* = 120) | *p* Value |
|---|---|---|---|
| Type of gastrectomy | | | 0.121 |
| Total gastrectomy | 34 (28.3%) | 42 (35.0%) | |
| Subtotal distal gastrectomy | 86 (71.7%) | 78 (65.0%) | |
| Type of reconstruction | | | 0.140 |
| Billroth 2, gastrojejunostomy | 21 (17.5%) | 29 (24.2%) | |
| Roux-en-Y (either gastric or esophagel) | 99 (82.5%) | 91 (75.8%) | |
| Combined resection | 4 (3.3%) | 6 (5.0%) | 0.093 |
| Gallbladder | 6 | 4 | |
| Spleen | 0 | 2 | |
| Distal pancreas | 0 | 1 | |
| Transverse colon | 1 | 2 | |
| Operative time (min.), Mean ± SD (Range) | | | |
| Overall | 212.3 ± 46.6 (120–400) | 192.4 ± 42.6 (120–320) | 0.012 |
| Subtotal gastrectomy | 180.9 ± 43.6 (120–360) | 180.2 ± 39.9 (180–290) | 0.560 |
| Total gastrectomy | 235.4 ± 45.7 (160–400) | 213.7 ± 38.8 (140–320) | <0.001 |
| Radicality, *n* (%) | | | 0.432 |
| R0 | 117 (97.5) | 115 (95.8) | |
| R1 | 3 (2.5) | 5 (4.2) | |
| Conversion to open, *n* (%) | 3 (2.5) | - | |

**Table 2.** *Cont.*

| Characteristics | LG (*n* = 120) | OG (*n* = 120) | *p* Value |
|---|---|---|---|
| Lauren classification (%) | | | 0.347 |
| Intestinal | 55 (45.8) | 59 (49.2) | |
| Diffuse | 46 (38.3) | 40 (33.3) | |
| Mixed | 15 (12.5) | 16 (13.3) | |
| Unknown | 4 (3.4) | 5 (4.2) | |
| Tumour differentiation (%) | | | 0.420 |
| Good | 23 (19.2) | 24 (20.0) | |
| Moderate | 45 (37.5) | 43 (35.8) | |
| Poor | 48 (40.0) | 50 (41.7) | |
| Undifferentiated or Signet ring cell carcinoma | 4 (3.3) | 3 (2.5) | |
| pT-stage, *n* (%) (UICC 8th ed.) | | | 1.000 |
| pT1 | 0 | 0 | |
| pT2 | 29 (24.1) | 29 (24.1) | |
| pT3 | 47 (39.2) | 47 (39.2) | |
| pT4a | 41 (34.2) | 41 (34.2) | |
| pT4b | 3 (2.5) | 3 (2.5) | |
| pN-stage, n (%) (UICC 8th ed.) | | | 0.987 |
| pN0 | 22 (18.3) | 20 (16.7) | |
| pN1 | 48 (40) | 41 (34.2) | |
| pN2 | 16 (13.4) | 11 (9.2) | |
| pN3a | 12 (10) | 29 (24.1) | |
| pN3b | 22 (18.3) | 19 (15.8) | |
| pStage, *n* (%) (UICC 8th ed.) | | | 0.798 |
| IIA | 42 (35.0) | 45 (37.5) | |
| IIB | 22 (18.3) | 10 (8.3) | |
| IIIA | 20 (16.7) | 21 (17.5) | |
| IIIB | 14 (11.7) | 26 (21.7) | |
| IIIC | 22 (18.3) | 18 (15.0) | |
| Clearance of surgical margin (negative) | | | |
| Overall | 117 (97.5%) | 115 (95.8%) | 0.432 |
| Proximal margin | 118 (98.3%) | 118 (98.3%) | 1.000 |
| Distal margin | 119 (99.2%) | 117 (97.5%) | 0.295 |
| Retrieved number of LNs, *n* | 31.4 ± 11.4 | 33.3 ± 15.4 | 0.134 |
| Mean ± SD (range) | (16–79) | (16–97) | |
| Number of positive lymph nodes, *n* (%) | 76 (63.3) | 80 (66.7) | 0.760 |

LG: laparoscopic assisted gastrectomy; OG: open gastrectomy; SD: standard deviation.

The preferred type of reconstruction was Roux-en-Y anastomosis, performed in 91 cases of patients in the LG and 78 in the OG, while Billroth II anastomosis was performed in the remaining cases, with no statistically significant difference ($p = 0.140$) between the two groups referring to these operative characteristics (Table 2).

Combined resections were performed on seven patients in the LG group and on nine patients in the RG group (3.3 vs. 5.0%, $p = 0.093$). The resected organs or tissues included the gallbladder (LG = 6 vs. OG = 4), spleen (LG = 0 vs. OG = 2), distal pancreas (LG = 0 vs. OG = 1), and partial transverse colon (LG = 1 vs. OG = 2).

Globally, the mean operative time was slightly longer in the LG group as compared to the OG group (212.3 ± 46.6 vs. 192.4 ± 42.6, $p = 0.012$). If we consider the subgroup analysis according to the type of gastrectomy, the length of the intervention of laparoscopic subtotal distal gastrectomy tended to approach the length of the open procedure (180.9 ± 43.6 in LG vs. 180.2 ± 39.9 in OG, $p = 0.560$), while the discrepancy was even more evident in the subgroup of total gastrectomy cases (235.4 ± 45.7 in LG vs. 213.7 ± 38.8 in OG, $p < 0.001$) (Table 2).

### 3.2. Pathologic Characteristics

There were no significant differences in tumor location, tumour differentiation, and Lauren's histotype between the two groups (Tables 1 and 2). As a result of the propensity score matching analysis, the pT stage was perfectly superimposable between the two groups ($p = 1.000$). The population predominantly consisted of patients with T3 and T4 tumors (73%). Almost one fourth of all patients had a T2 depth of invasion. Four patients

with T4b tumours had required associated organ resection, consisting of transverse colon resection (one in LG and two in OG groups) and one distal pancreasectomy (OG group) for direct cancer infiltration.

Globally, no difference in the mean number of harvested lymph nodes ($31.4 \pm 11.4$ vs. $33.3 \pm 15.4$; $p = 0.134$) and in tumor node metastasis (63% in the LG group, compared with 67% in the OG group; $p = 0.760$) was observed between the LG and OG group. As could therefore be expected because they had been matched for tumour invasion depth (pT), no statistically significant differences in the extent of nodal invasion (pN) were found ($p = 0.987$, Table 2). Specifically, in the LG group, 22, 48, 16, 12, and 22 patients had N0, N1, N2, N3a, and N3b nodal invasion, respectively. In the OG group, 20, 41, 11, 29, and 19 patients had N0, N1, N2, N3a, and N3b nodal invasion, respectively. Thus, even though the pT stage was perfectly aligned as result of the propensity matching method, a modest and not statistically significant discrepancy in the extent of lymph node infiltration was found among patients. This can be attributed to some aspects responsible for the different lymph node spread despite the same tumour depth of invasion, such as biological aggressiveness, peritumor lympho-vascular infiltration, and/or other not yet scientifically known tumour features.

Pathological stages according to the 8th UICC classification were as follows: LG group had 42 (35.0%) cases at stage IIA, 22 (18.3%) at stage IIB, 20 (16.7%) at stage IIIA, 14 (11.7%) at stage IIIB, and 22 (18.3%) at stage IIIC; whereas the OG group had 45 (37.5%) cases at stage IIA, 10 (8.3%) at stage IIB, 21 (17.5%) at stage IIIA, 26 (21.7%) at stage IIIB, and 18 (15.0%) at stage IIIC. There were no significant differences in tumour stage distribution between the two study groups (Table 2).

Eight patients (3.3%) had microscopic cancer cells in the margin of the specimen, consistent with an R1 resection; all other patients received an R0 resection. Margin clearance was satisfactory, ranging from 97% to 99% in the laparoscopic group and from 95% to 97% in the open group, and the *p* value showed no significant difference between the two groups in the rate of either proximal (LG 1.7.% vs. OG 1.7%; $p = 1.000$) or distal (LG 0.8% vs. OG 2.5%; $p = 0.295$) resection margin involvement

### 3.3. Post-Operative Outcomes

Table 3 summarizes the postoperative adverse events. The all-grade overall complication rate within 30 postoperative days was 18.7%. The postoperative morbidity rate was 16.7% (20 of 120 patients) in the LG group and 20.8% (25 of 120 patients) in the OG group, with a trend in favour of the laparoscopic group but not statistically significant ($p = 0.210$).

There were no significant differences between the two groups with regard to the incidences of each subtype of complication, except for pulmonary infection and wound infections, which occurred less frequently after LG (0.8% vs. 4.2%, $p = 0.001$ for the two categories).

No significant differences were found in the stratification of the diverse grade of Clavien–Dindo complications, except for Grade II pulmonary infection and Grade III wound complication, which occurred more frequent in the OG group (1 vs. 4 and 0 vs. 3, $p < 0.05$). Specifically, in the laparoscopic group, one patient experienced Clavien–Dindo Grade II complication for wound infection (opened at bed side, but needing modification of therapeutic regimen with antibiotics), one patient experienced Clavien–Dindo Grade II for lung infection (requiring pharmacological treatment with drugs, i.e., antibiotics, other than such allowed for grade I complications, i.e., antiemetics, antipyretics, analgesics, diuretics, and electrolytes); in the open group, two patients experienced Clavien–Dindo Grade II complication for wound infection, three patients experienced Clavien–Dindo Grade III complication for wound infection (which included wound dehiscence requiring surgical intervention, two not under general anesthesia, Grade IIIa, and one under general anesthesia—Grade IIIb), four patients experienced Clavien–Dindo Grade II for lung infection and one had Clavien–Dindo grade III (lung infection needing closed drainage of the pleural cavity not under general anesthesia—Grade IIIa).

**Table 3.** Adverse events.

| Variables | LG (*n* = 120) | OG (*n* = 120) | Odds Ratio (RR 95% CI) | *p* Value |
|---|---|---|---|---|
| Postoperative complications, *n* (%) | | | | |
| Overall | 20 (16.7) | 25 (20.8) | 1.3 (−1.9 to 6.6) | 0.210 |
| Anastomotic leakage (1) | 4 (3.3) | 3 (2.5) | 2.3 (−1.1 to 3.9) | 0.500 |
| Duodenal stump leakage (2) | 3 (2.5) | 1 (0.8) | 0.4 (−1.2 to 2.1) | 0.142 |
| Leakages (1) + (2) | 7 (5.8) | 4 (3.3) | 1.6 (−1.1 to 4.2) | 0.072 |
| Anastomotic bleeding | 1 (0.8) | 1 (0.8) | 0.2 (−1.1 to 1.1) | 1.000 |
| Pancreatitis/Pancreatic fistula | 1 (0.8) | 2 (1.7) | 1.3 (−0.1to 2.0) | 0.681 |
| Intra-abdominal collection | 2 (1.7) | 2 (1.7) | 0.4 (−1.4 to 1.4) | 1.000 |
| Intra-abdominal bleeding | 2 (1.7) | 2 (1.7) | 0.4 (−1.4 to 1.4) | 1.000 |
| Bleeding wound complication | 0 | 1 (0.8) | −0.2 (−0.6 to 1.2) | 0.446 |
| Wound infection | 1 (0.8) | 5 (4.2) | 0.8 (−1.1 to 4.3) | <0.01 |
| Pulmonary effusion | 1 (0.8) | 1 (0.8) | −0.2 (−1.1 to 1.1) | 1.000 |
| Pulmonary infection | 1 (0.8) | 5 (4.2) | −0.8 (−2.2 to 0.3) | <0.01 |
| Intestinal obstruction/ileus | 2 (1.7) | 2 (1.7) | −1.1 (−1.4 to 1.4) | 0.475 |
| Cardiac | 1 (0.8) | 0 | 0.2 (−0.6 to 1.2) | 0.695 |
| Hepatic | 1 (0.8) | 0 | 0.2 (−0.6 to 1.2) | 0.744 |
| Reoperation, *n* (%) | 4 (3.3) | 3 (2.5) | | 0.714 |
| Clavien–Dindo classification | | | | |
| Grade II (%) | 7 (5.8) | 11 (9.2) | | 0.281 |
| Wound infection | 1 | 2 | | 0.740 |
| Pulmonary infection | 1 | 4 | | <0.05 |
| Anastomosis bleeding | 1 | 1 | | 1.000 |
| Pancreatitis | 1 | 0 | | 1.000 |
| Bleeding wound complication | 0 | 1 | | 1.000 |
| Intra-abdominal collection | 1 | 1 | | 1.000 |
| Pulmonary effusion | 1 | 1 | | 1.000 |
| Intestinal obstruction/ileus | 1 | 1 | | 1.000 |
| Grade III (%) | 7 (5.8) | 10 (8.3) | | 0.628 |
| Wound infection | 0 | 3 | | <0.05 |
| Pulmonary infection | 0 | 1 | | 1.000 |
| Intra-abdominal collection | 1 | 1 | | 1.000 |
| Anastomotic leakage | 2 | 1 | | 1.000 |
| Duodenal stump leakage | 1 | 1 | | 1.000 |
| Intestinal obstruction/ileus | 1 | 1 | | 1.000 |
| Intra-abdominal bleeding | 2 | 2 | | 1.000 |
| Grade IV (%) | 1 (0.8) | 0 | | 1.000 |
| Anastomotic leakage | 1 | 0 | | |
| Grade V (%) | 5 (4.2) | 4 (3.3) | | 0.453 |
| Anastomotic leakage | 2 | 2 | | 1.000 |
| Heart failure | 1 | 0 | | 1.000 |
| Pancreatitis/Pancreatic fistula | 0 | 2 | | 1.000 |
| Duodenal stump leakage | 2 | 0 | | 1.000 |
| Clavien–Dindo grade ≥III (%) | 13 (10.8) | 14 (11.7) | | 0.625 |
| Mortality, *n* (%) | 5 (4.2) | 4 (3.3) | | 0.987 |
| Postoperative hospital stay (days; mean ± SD) | 9.1 ± 4.7 | 11.6 ± 4.8 | | <0.001 |

LG: laparoscopic gastrectomy; OG: open gastrectomy; CI: confidence interval; RR: relative risk.

The anastomotic and duodenal stump leakage rate tended to be globally higher in the LG group compared to the OG group (5.8% vs. 3.3%), but this was not statistically significantly different (*p* = 0.072). Specifically, in the laparoscopic group, two patients experienced Clavien–Dindo Grade III complication for surgical leakage (one duodenal stump and one gastro–jejunal), one patient experienced Clavien–Dindo Grade IV complication for esophageal–jejunal anastomosis leakage (requiring ICU—Intensive care unit—management), four patients died within 30 days of in-hospital course (Clavien–Dindo Grade V) for septic complications subsequent to (*n* = 2) esophageal–jejunal anastomosis leakage and (*n* = 2) duodenal stump leakage; in the open group, two patients experienced

Clavien–Dindo Grade III complication (one duodenal stump and one esophageal–jejunal anastomosis leakage), and two patients died from Clavien–Dindo Grade V septic complications subsequent to esophageal–jejunal anastomosis leakage.

Globally, Clavien–Dindo Grade II-III complications were the most frequent. No significant differences were revealed for major complications (Clavien–Dindo Grade $\geq$ III) between the two groups (10.8 vs. 11.7%; $p$ = 0.625).

The patients treated by laparoscopic technique developed higher rates of re-interventions, which were not statistically significant (3.3% vs. 2.5%, $p$ = 0.714).

Post-operative mortality within 30 days was 4.2% in the LG group ($n$ = five patients, four due to septic complications subsequent to anastomotic or duodenal stump leakage, one due to heart failure) and 3.3% in the OG group ($n$ = four patients, two because of septic complications after anastomotic leakage, and the other two from multiorgan failure subsequent to pancreatitis and pancreatic fistula) ($p$ = not significant) (Table 3).

The post-operative hospital stay was 9.1 $\pm$ 4.7 days in the laparoscopic group, which was significantly less than that in the open group (11.6 $\pm$ 4.8 days, $p$ < 0.001).

## 4. Discussion

Laparoscopic gastrectomy with lymph node dissection has developed as a minimally invasive surgery for gastric cancer over the past 25 years. This surgery has been used mainly for early-stage gastric cancer. Sufficient data are available on the feasibility of LADG, and this approach has essentially been validated for early gastric cancer, as several level III studies and meta-analysis demonstrated that laparoscopic gastrectomy with limited lymphadenectomy for patients with early gastric cancer had non-inferior oncologic outcome relative to open surgery, with better short-term results [7,8,12,13,16,40–42]. The potential benefits of LG compared to conventional open surgery include faster postoperative recovery, quicker return of gastrointestinal function, shorter hospital stay, less postoperative pain, and better cosmetics.

On the other hand, complete lymph node dissection is a crucial element for curative surgical treatment for AGC, and radical gastrectomy with D2 lymph node dissection is currently considered the standard surgical approach for resectable locally advanced cancer patients [1–3]. With respect to early stage gastric cancer, fewer large well-conducted series concerning the safety of laparoscopic-assisted distal and total gastrectomy had been focused on AGC. One of the biggest obstacles is the more difficult surgical procedure involved in the laparoscopic operation for advanced gastric cancer, specifically extended (D2) lymphadenectomy and reconstruction of the digestive tract, which require elevated skills in laparoscopic procedures. Along with these reasons, the low acceptance of LG in gastric cancer is due to the scarce evidence regarding the oncological adequacy of laparoscopic procedures, in particular whether they can achieve the same effect in D2 lymphadenectomy as open surgery, as well as the long-term results.

The majority of the comparative trials between LG and the traditional open technique for the treatment of AGC are retrospective limited single-centre studies and are often too heterogeneous to be globally evaluated [22,43–55]. Only a few Phase III randomized controlled trials have been prospectively conducted as comparative studies [56–61].

If individually taken, most of the studies are too small to reach the necessary statistical power to draw definitive conclusions, and the majority of these contain a greater proportion of patients operated on for early distal gastric cancer, thus making it implausible to obtain results generalized to all gastric cancer stages. Even when focused on only AGC, the main reasons for the heterogeneity of the studies available so far have been the different levels of laparoscopic expertise, the issue related to the learning curve, mixed gastrectomy types (with the prevalence of LADG), different levels of lymphadenectomy, sometimes a lack of randomization, nonblinded assessment of outcomes, and a predominance of Asian studies.

Several meta-analyses [62–75] have been conducted to enhance the statistical power of data, trying to overcome the lack of the breadth of the series focused on AGC, but definitive conclusions are not simple to draw, and some controversy on the potential superiority of LG

over traditional procedure still persists. In fact, most of the studies included in these meta-analyses are non-randomized comparative trials, and a high heterogeneity exists between them, which could significantly affect the final results of pooled data. Even a Cochrane systematic review published in 2016, including RCTs comparing laparoscopic versus open gastrectomy for gastric cancer, considered that the quality of evidence of comparative data was low, with a significant share of bias in the identified studies, suggesting that the superiority of laparoscopic gastrectomy over traditional surgery could not be asserted with certainty [76].

However, recently, important large, high-quality eastern RCTs, whose results were eagerly awaited, have been published [28,56,58–61,77,78]. Among the most significant, the Korean Laparoscopic Surgical Society (KLASS) group launched the multi-centre RCT KLASS-02 [77] in 2015, with the aim of comparing the oncologic and surgical outcomes of LADG with D2 lymphadenectomy for patients who were clinically diagnosed with locally AGC, with respect to conventional open subtotal gastrectomy and D2 lymphadenectomy. The results of this study are now available and they have demonstrated benefits in terms of short-term outcomes (lower complication rate, faster recovery, less pain compared with open surgery) [61] and a comparable 3-year relapse-free survival rate with respect to ODG [78]. Similar multi-institutional prospective RCTs comparing LADG with ODG for locally AGC were conducted in Korea (COACT1001 trial) [56]. This study showed no significant difference in the overall postoperative complication rate between LADG vs. ODG, in addition to an identical compliance rate of D2 lymph node dissection, thus confirming more than the safety the oncologic adequacy of LADG. Similar to the Korean group, the Chinese Laparoscopic Gastrointestinal Surgery Study (CLASS) Group conducted an analogously designed multicentre RCT study (CLASS-01) [58] over the same recent time period: 1.056 patients with AGC were randomized in 14 Chinese institutions between open and laparoscopic distal gastrectomy. A D2 lymphadenectomy was required, with mean numbers of harvested lymph nodes of 36.1 vs. 36.9 (not significantly different) in the respective laparoscopic and open groups. They confirmed no inferior short-term outcomes of LDG compared to those of ODG in the treatment of patients suffering from AGC [58], with equal 3-year disease-free survival [28].

Unfortunately, most of these recent RCTs are subsequent to the above-mentioned meta-analysis and Cochrane review, which therefore could not include them. Only the two latest meta-analyses are recent enough to include the above-mentioned RCTs [71,72], and they confirmed that LG with D2 lymphadenectomy is a safe procedure and can be performed with equivalent overall short-term morbidity and mortality versus the open approach for patients with AGC

Globally, the recent high-quality RCTs [28,56,58–61,77,78] confirmed the essentially favorable short-term outcomes of LG over OG, such as a faster time of bowel canalization and oral intake, lower blood loss, and shorter length of hospital stay, despite a longer duration of surgery, at the same time as comparable post-operative complication rates and an analogous number of dissected lymph node. Thanks to results shown by these high-quality large multicentre Eastern RCTs, we are seeing a constantly increasing shift in the indication of LG from early gastric cancer, passing through Stage I disease, to locally advanced cancers. In 2014, the Japanese gastric cancer treatment guidelines (ver. 4) introduced the laparoscopic distal gastrectomy as one treatment option in general practice for stage I gastric cancer [79], and it is not unlikely that in the near future the international guidelines will be updated on the basis of new evidence on AGC.

However, the early detection screen for gastric cancer has not been in widespread use in the majority of countries, and so unfortunately most patients have been diagnosed at advanced stages, and the application of LG for AGC remains debatable. Moreover, detailed analyses indicate slight discrepancies among the recent RCTs regardless of the country of origin. They may be more plausibly influenced by different strategies among the trials and by inter-group/inter institutional variability. For example, the trials reported by Hu et al. [58], Park et al. [56], and Wang et al. [60] included patients with clinical tu-

mor stages cT2 to 4a, N0 to N3, and cM0, submitted only to LADG, whereas the trial reported by Shi et al. [59] excluded patients with a T-stage higher than T3 and also included patients with planned, other than distal, total, and proximal gastrectomy. In the trial of Shi et al. [59], anastomosis was conducted by stapling, whereas the remaining trials did not provide information on anastomosis. As the anastomotic technique is particularly challenging in laparoscopic total gastrectomy (LTG), the number of anastomosis complications may be different depending on the technique used. While the trial reported by Wang et al. [60] included only surgeons who had performed at least 60 laparoscopic and 60 open gastrectomies, surgeons had to have performed at least 50 distal gastrectomies with D2 lymphadenectomy using open and laparoscopic approaches in the trials reported by Hu et al. [58] and Shi et al. [59], and the trial reported by Park et al. [56] required only 30 laparoscopic distal gastrectomies.

In the Chinese RCT of Hu et al. [58], the morbidity rate in the laparoscopic group (15.2%) was, although statistically not significant, slightly higher than that of the open group (12.9%), and in turn the latter was almost half of that in KLASS-02-RCT [61]. In fact, in the KLASS-02-RCT, the early morbidity rate was significantly lower after LDG (16.6%) than after ODG (24.1%; $p = 0.003$). In an intermediate plane between the two, Wang et al. [60] and Shi et al. [59] reported a higher complication rate in the open vs. laparoscopic group, but this was not statistically significant (LADG 13.1% vs. ODG 17.7%, $p = 0.174$, and 11.72% LAG vs. 14.38% OG, $p = 0.512$, respectively).

In terms of radicality of surgery, Chinese RCTs showed a lower number of retrieved lymph nodes (Wang et al. [60] 29.5 in laparoscopy vs. 31.4 in open; Hu et al. [58] 36.1 vs. 36.9; Shi et al. [59] 31.5 vs. 32.1) than KLASS-02-RCT [61] (46.6 vs. 47.4).

Therefore, despite disputes and that the design of our study is not comparable with those previously reported, by being retrospective and lacking in randomization, we may overcome some of the above described discrepancies. Our study's strength lies in the rigorous comparative analysis, made possible through a strict propensity score estimation and matching, which allowed the most homogeneous possible comparison between LG and OG. Such a method, designed to reduce the effect of selection bias and to balance the baseline covariates between the study groups, may overcome some of the biased estimates of treatment effects in the observational study. Appropriate adjustment for potential confounding factors is the correct method to adjust for significant differences in the patients' clinicopathologic characteristics, and it is essential in evaluating the effectiveness of an intervention, particularly when the study group is heterogeneous. Studies employing a propensity score matching method allow the congruity of data and limit the bias resulting from the heterogeneity of the participants, rather than broadening the cohort series through prospective randomization, representing a notable methodology for such confounder adjustment and aiming to approach the quality of an RCT [39,80]. An ever increasing number of researchers have resorted over time to retrospective studies using such a method, and their results are essentially in line with those presented here [26,81–85].

Specifically, our relatively low number of patients compared to the numbers usually, although not always, achieved by randomized trials, is balanced by the match control method. Through restriction to a limited category of patients, which comprise a uniform type of surgical procedure (in our case, radical gastrectomy with D2 lymphadenectomy), and minimizing the difference in patients' characteristics by matching the covariates, the flaw deriving from the lack of prospective randomization is virtually offset.

Another advantage of this study, being retrospective, is to permit a stringent comparison of homogeneous data and to exactly align post-operative tumour characteristics, such as the pT stage. This makes it possible to overcome those discrepancies derived from an inevitable rate of the pre-operative diagnostic inaccuracy, which generates differences between pre-operative (cStage) and post-operative (pStage) findings. This allowed us to leave out from the study analysis over- and under-staged patient data, attempting to come closer to the truth of the tumour's characteristics, which instead is inevitably burdened by a certain amount of discrepancy in observational studies.

The current study has the privilege of investigating the safety and feasibility of LG in a Western population, frequently presenting multiple comorbidities and undergoing total gastrectomy in a significant proportion. In fact, one of the limitations of the previous Eastern RCTs is that their eligibility criteria permit the inclusion of mainly patients requiring distal gastrectomy. This may be unsatisfactory for countries other than East Asia, as proximally located advanced cancer requiring total gastrectomy, which involves more challenging and complex procedures than distal gastrectomy, is predominant in the West [86,87]. The applicability of Eastern evidence to Western countries remains not so linear, as general and intrinsic gastric cancer population differences do in fact exist. Our under-study population predominantly consisted of patients with T3 and T4 tumors, representing the stage at which most Western patients present with gastric cancer. The lower overall incidence of gastric cancer in the West, in spite of the greater proportion of younger patients with advanced stage and upper third cancer (requiring total gastrectomy), together with diverse gastric tumor molecular biological aggressiveness, etiology, genetic arrangement, and geographic spread with respect to Eastern countries, is well-known [87–89]. Moreover, we only included cases who underwent gastrectomy with D2 lymph node dissection, which has been accepted as the standard for AGC. Granting a Western experience in this specific procedure conducted laparoscopically, whose effectiveness still needs a definite validation, makes our study a worthy contribution to literature knowledge.

Hence, while keeping in mind the above-mentioned differences and considering that the present study included only patients who had pathological diagnosis of local AGC, our results are substantially in line with those previously reported in the literature. Previous historical unmatched patient results from specialized centres with sufficient experience of open D2 lymph node dissection for advanced gastric cancer reported morbidity rates of 20.1–33.5% and mortality rates ranging from 0.8 to 3.1% [90–93]. Similarly, we revealed acceptable and comparable postoperative 30-day overall complication rates (overall morbidity rates 16.7% and 20.8% in the LG and OG groups respectively, with no significant difference, $p = 0.210$) and mortality rates (4.2% and 3.3% in the LG and OG groups respectively, $p = 0.987$). No significant difference in the major complication rates was revealed (Clavien–Dindo Grade $\geq$ III: 10.8 and 11.7%, respectively $p = 0.625$).

Conversely, although the overall postoperative morbidity was not statistically different, the trend in favour of the laparoscopic procedure was significant in the subgroup analysis of some types of complications. Specifically, a statistically significant inferior rate of pulmonary infection (overall 0.8% in LG vs. 4.2% in OG, $p < 0.01$; Clavien–Dindo Grade II 0.8% in LG vs. 3.3% in OG, $p < 0.05$) and a lesser degrees of surgical site infection (overall 0.8% in LG vs. 4.2% in OG, $p < 0.01$; Clavien–Dindo Grade III 0% in LG vs. 2.5% in OG, $p < 0.05$) were revealed in the LG group.

From these viewpoints, LG could be technically feasible with better short-term outcomes, and these results are consistent with previously published studies [12,57,65,94,95]. Certainly, part of the reason for this evidence can be explained by the intrinsic nature of the minimally invasive procedure, which allows minor surgical trauma and less immunosuppression than open surgery, as are widely reported in literature and well recognized as being among the common benefits of the laparoscopic approach. This is consistent with other advantages of LG over open surgery shown by several other reports, which have demonstrated less surgical trauma, reduced blood loss, less postoperative pain, and earlier return to normal bowel function in LG [18,64,96–98]. Specifically with regard to pulmonary complications, the systematic review and meta-analysis of laparoscopy-assisted and open total gastrectomy for gastric cancer conducted by Chen K et al. [98] revealed a significant decrease in medical complications ($p = 0.03$) in favour of the minimally invasive procedure, with a possible contribution from respiratory complications, which occurred less often, although not statistically significant ($p = 0.06$), than in the open technique. That could be attributed to the reduced invasiveness of the laparoscopic technique and less postoperative pain. The reduced length of the incision wound compared to laparotomy, with consequent less pain due to the minor presence of tension sutures, can improve the post-operative

patient's respiratory dynamics and ability to perform exercise breathing, thus leading to fewer complications such as pulmonary infection [98].

To further corroborate this phenomenon, a lower rate of grade I pulmonary complications has been reported in favour of total laparoscopic total gastrectomy with respect to laparoscopy-assisted total gastrectomy [99]. The larger and closer to epigastrium mini-laparotomy incision of the laparoscopic-assisted technique compared to that totally laparoscopic may explain, together with corresponding worse pain score, decreased pulmonary function in the former group due to the limited movement of the diaphragm and deep breathing, with a consequent higher pulmonary complication rate.

Presumably for the same reason, the laparoscopic procedure is associated with a shorter postoperative hospital stay as well, despite a longer operation time, with respect to OG (in our series 9.1 vs. 11.6 days, $p < 0.001$). Indeed, the significantly longer operative time for the laparoscopy group, as shown in other articles [20,50,100], did not translate into an increased perioperative complication rate. Consistently, the operation time in our experience was longer (LG 212.3 vs. OG 192.4 min, $p = 0.012$). Our prolonged operation time might be attributable to a substantial number of total gastrectomies and the low conversion rate. In the present study, the assumed gap of operative times between LG and OG was greater in total gastrectomies, while the duration of laparoscopic distal gastrectomy tended to approach to that for laparotomy (Table 2). This implied that total gastrectomy remains a challenging and time-consuming procedure, particularly in laparoscopy. This gap is expected to decrease in the near future thanks to improvements in laparoscopic techniques and advances in the team's learning curve.

On the other hand, we revealed a percentage of duodenal stump and anastomosis leakages that, even if statistically comparable between the two groups, was demonstrated to be basically higher in the laparoscopic procedure. We had seven (5.8%) patients with this type of complication in LG, with four deaths due to consecutive septic aggravation, while it was 3.3% in the OG group (4/120 patients, with two consequent deaths). Conversely, we did not found anastomotic structure complication, probably depending on the latency of this type of complication, which makes it impossible to detect by such a short-term (30-days post-operative) investigation. In addition, an eastern RCT showed that anastomotic leakage tended to be more frequent in laparoscopic distal gastrectomy than in open distal gastrectomy in patients with AGC [58]. Nevertheless, the percentages of anastomotic leakage in the current study, together with the overall mortality rates (4.2% in LG and 3.3% in OG), tended to be higher than those reported in previous high-quality Asian trials (anastomotic and duodenal stump leakage ranging between 1.4% and 3.7% in LG and 0% and 2.0% in OG; post-operative mortality rate ranging between 0.0% and 0.4% in LG and 0.0% and 1.0% in OG) [55–61,78,101,102], but they are similar to or lower than those reported by Western research. For example, Huscher et al. [103] reported an incidence of anastomotic and duodenal stump leakages of 11% in patients submitted to laparoscopic gastrectomy, Moisan et al. [104] reported 12.9% and 6.4% in the respective laparoscopic and open groups, and Orsenigo et al. [105] reported 18.3% and 5.2%, respectively. Another recent Italian propensity score-matched case-control study [106], which compared robotic gastrectomy vs. laparoscopic gastrectomy vs. open gastrectomy, revealed lesser rates of anastomotic leakage (2.6% in the laparoscopic group, 2.6% in the robotic group, and 3.6% in the open group, with no statistical difference between the groups, $p = 0.78$), but this study's cohort also included a predominant share of early-stage gastric cancer patients (56.3%). With regards to mortality, our results are similar to the Western studies: Husher et al. [103]: 7% of laparoscopic gastrectomies, and successively in their RCT of 2005 [43] reported mortality rates of 3.3% and 6.7% in the laparoscopic and open group, respectively. Orsenigo et al. [104] reported a lower percentage of mortality, 2.0% and 1.4%, respectively, in LG and OG. Two recently published European multicentre RCTs evaluating the effectiveness of laparoscopic vs. open gastrectomy for gastric cancer reported similar outcomes: the LOGICA trial [107] showed rates of 8.7% and 10.0% of anastomotic leakage in the laparoscopic and open group, respectively, and 4.3% vs. 6.4% for 30-day postoperative mortality,

respectively. In the STOMACH trial [108], the rates of anastomotic leakages were 8.5% in the minimally invasive group and 10.2% in the open group, while no 30-day postoperative mortality information was reported.

The notoriously long learning curve of LG is probably responsible for a great deal of the above, but in part it is certainly due to the fact that the Western population consists of older patients with more advanced tumors, multiple comorbidities, and higher BMI [87–89], and the Western patient more frequently undergoes total gastrectomy [109], a blatantly more difficult surgery, than the general Asian patient.

Anastomotic complications can be affected by the surgeon's experience [110]. In our study, it was not easy to weigh the impact of the different learning phases on the potential increase in complications, as each operating surgeon was introduced at different times to the team who performed LG. However, it can be stated that certainly in the present analysis, operator-related bias was minimized because both LG and OG were performed by board-certified surgeons who had had adequate experience as operators (minimum experience level of 40 gastrectomies), and all laparoscopic procedures were performed or supervised by the most experienced surgeon in charge. Moreover, our institutes can reasonably be considered highly proficient in oncologic gastric surgery, as they cross the line of 21 procedures performed per year, which has been associated with better survival results in gastric cancer patients [29]. On the other hand, the learning curve for LG is not well defined in the literature, ranging from 20 up to 100 cases, and the minimum number of cases to define a high-volume center for gastric surgery is not equally delineated all over the world (in Italy it is about 25–40 cases per year, according to the Italian Ministry of Health (www.oncoguida.it) and "Programma Nazionale Esiti (PNE)" [111], but no cut-off number of procedures per single surgeon has been established.

Regarding the evaluation of oncologic effectiveness, our histopathological results are comparable between the two groups; in particular, LG was equal to OG in the D2 lymph node dissection, radicality (R0), and clearance of margin. As the most objective index of lymphadenectomy is the comparison of the number of lymph nodes obtained between the LG and OG groups, certainly many researchers [19,22,45,112–114] have reported a similar number of lymph nodes resected in LG and OG with D2 dissection for AGC, thus demonstrating that the laparoscopic procedure is oncologically adequate. Based on our experience, laparoscopic procedures offer unparalleled amplified clarity for identification of anatomical structures, which is particularly useful to enable the precision isolation of lymph nodes while performing dissection and sealing in extensive lymphadenectomies. That surely depends in part on the skill level obtained by the surgeons and also to the technical progress that has been achieved in recent years, such as advances in optics and improvements of the latest laparoscopic energy devices. This probably balances the complex maneuverability of laparoscopic instruments and overcomes some of the notorious drawbacks of laparoscopic procedures, which in the past have been associated particularly with limitations in the ability to perform extended lymphadenectomies [42,62,115–118]. In fact, with the passing of time, an increasing number of surgeons have demonstrated their ability to perform total gastrectomy and adequate laparoscopic D2 lymphadenectomy [19–21,81].

To summarize, we confirm that LG is a feasible and safe procedure treating locally advanced gastric cancer, showing perioperative results similar to those obtained with the standard open technique and ensuring, at the same time, adequate oncologic equivalency. The laparoscopic procedure was shown to be able to maintain the well-known benefits of its minimally invasive nature (minimal trauma and quick recovery), which decreases the length of the hospital stay, thus corroborating the effectiveness of the minimally invasive approach already well known in the literature. These findings are validated in the present study by the propensity score match analysis. The advantage of our study is to show the experience with laparoscopic gastrectomy in a Western European population of predominantly advanced gastric adenocarcinomas, thus including a certain proportion of total gastrectomy, which requires more challenging procedures and advanced surgical skill.

This report could serve as a background for a future study on the long-term survival and oncologic efficacy of laparoscopic surgery in AGC.

The limitations of the present study include the recruitment of a relatively small sample size of participants compared to recent larger RCTs. Second, the study findings may be limited by the retrospective nature, although the dataset was prospectively collected and a strict matching of the potential confounding factors was conducted. We included all consecutive patients who met the inclusion criteria to avoid any selective bias, and it was also verified that clinical–pathological characteristics (age, gender, BMI, comorbidity, ASA, adjuvant therapy, tumour location, type of gastrectomy, pT stage) were well balanced between the two groups by the propensity matched-method. Nevertheless, there is no guarantee that all confounding factors could be offset by the propensity score matching method in our analyses, and unmeasured or unobserved effects of inherent selection biases could not be completely eliminated by using this method. For example, potential heterogeneity may derive from some discrepancies between the two teams involved in the two centres. The different level of expertise achieved by the surgeons (learning curve) and differences in surgical preferences (such as anastomosis and reconstruction types), different habits of the operating room staff, differences in the surgical tools used, some potential discrepancies in the practices of the perioperative management of the patient (such as the detection point of complications, management of complications, and criteria for discharge), and last but not least the different pathologists who analyzed the tumoural specimen. All that might introduce intrinsic bias. However, it must be said that the two institutions in which this work was realized belong to the same Health Company (Central Tuscany Local Health Company, Italy), and thus the perioperative patient's management followed the same agreed standard protocols, and the surgical procedure has been homologous. Patients received the same standardized perioperative care with respect to the postoperative diet, fluid administration, pain control, and hospital discharge, regardless of the operation type, according to the ERAS (Enhanced Recovery After Surgery) program, which was formally introduced for gastric cancer patients before 2015 in one of our institutes (Prato) and in 2018 in the other one (Florence). We believe that, at least in part, this may have eliminated potential disparities between the two study centres in the postoperative management of patients. The collection of data was strictly carried out using the same criteria in the two centres. Finally, regarding potential discrepancies deriving from the different level of learning curve reached by the teams of the two centres, it is true that the database does not include surgeon identifiers, so we were unable to assess the relation between surgeon volume and patients' clinical outcomes. Nevertheless, in both institutions, the expertise was founded on highly specialized competency in oncological surgery and proficiency in open and in laparoscopic procedure, guaranteed by the presence for all included cases of an operator surgeon or at least a supervisor with a minimum of 40 procedures performed (for each laparoscopic and open technique).

Another limitation is that we did not perform cytological examination of peritoneal washes, even if we included T4a-4b tumours (serosa involved). That could be potentially related to a greater risk of peritoneal recurrence, but it should be elucidated by a long-term survey.

Thanks to the development of the laparoscopic technique, the use of LG for treating gastric cancer has expanded in the historically poorly proficient West, such as in the United States, Europe, and other countries [17,88,107,108,119], but presently the amount of literature evidence in western populations is still scarce. The results of other high-quality large ongoing studies (such as, in Japan JLSSG 0901 Phase III RCT, Registered Number: UMIN 000003420, www.umin.ac.jp/ctr/) are being awaited, but until definitive evidence is obtained, doubts still persist on the routine use of LG for AGC.

## 5. Conclusions

Despite the nonrandomized retrospective setting, our study confirms that LG with D2 lymph node dissection is a safe and feasible procedure in treating patients with locally

advanced gastric adenocarcinoma, with similar short-term results and oncologic radicality compared to the traditional open surgery. Moreover, the laparoscopic approach may provide the benefit of reducing some complications, such as pulmonary infections and wound complications, and faster recovery compared with the open technique. Beyond the contribution of experiences such as this, it would seem reasonable to affirm that the laparoscopic technique is advantageous with respect to the traditional open technique in terms of general early post-operative outcomes and recovery of normal activities for gastric cancer.

As Western literature evidence increases over time, it is plausible to believe that there will be increasing acceptance of LG for AGC, including the application of laparoscopic total gastrectomy, which is a notoriously challenging procedure with potentially non-negligible severe complications. Further well-designed, larger scale, prospective randomized studies are necessary to definitely assess the possible advantages of the laparoscopic approach in the treatment of AGC, especially in the West and with regards to long-term outcomes.

**Author Contributions:** S.C. (Stefano Caruso) and M.S. wrote and conceived the design of the study; M.M. and R.G. contributed to the literature search and acquisition of data; S.C. (Stefano Cantafio) and G.M.P. contributed to the critical appraisal of the work, revising the article critically for important intellectual content and supervising the interpretation of data. All authors have read and agreed to the published version of the manuscript.

**Funding:** This research received no external funding.

**Institutional Review Board Statement:** The study was conducted according to the guidelines of the Declaration of Helsinki and approved by Clinical Ethics Committee (COMEC—Comitato per l'Etica Clinica, AUSL—Azienda Unità Sanitaria Locale—Toscana Centro (protocol code 418/2013).

**Informed Consent Statement:** Informed consent was obtained from all subjects involved in the study.

**Data Availability Statement:** The data presented in this study are available on request from the corresponding author.

**Conflicts of Interest:** All authors disclose any potential or actual personal, political or financial conflict of interest in the material, information or techniques described in the paper.

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
