# Peer review of "Laparoscopic vs. Open Gastrectomy for Locally Advanced Gastric Cancer: A Propensity Score-Matched Retrospective Case-Control Study"

_curroncol, doi:10.3390/curroncol29030151_

Round 1

Reviewer 1 Report

I have no additional questions to the authors of the article

Good evening! Laparoscopic technologies have been used in gastric cancer
surgery for about 20 years. However, the feasibility of performing laparoscopic
gastrectomy continues to be actively discussed. The opinions of European and
Asian surgeons about its expediency differ, especially in patients with stage
II-III gastric cancer. The authors presented a retrospective analysis of
surgical treatment of 240 patients, of which 120 were patients with
laparoscopic gastrectomy and 120 with open surgery. Treatment was carried
out in 2 European clinics. The groups are comparable in terms of the main
clinical and morphological parameters. According to the analysis of treatment
results, no significant differences in the frequency of postoperative
complications and mortality were found. Despite the longer time to perform
the surgical stage of treatment, the terms of hospitalization of patients
with laparoscopic gastrectomy were less than 9.1 and 11.6 days. It is important
that the same number of perigastric lymph nodes were removed during laparoscopic
and open gastrectomy - 31.4 (16-79) and 33.3 (16-97), since the quality of
perigastric lymph node dissection determines the optimal long-term results
of treatment. The article provides a full discussion of the results obtained,
using modern literature on the problem of laparoscopic gastrectomy. The list
of cited articles is large - 117 sources. I believe that the article is of
great interest to practicing surgeons. I recommend it for publication. In the
future, I would like to see a comparative analysis of the overall and
relapse-free survival of these patients.

Reviewer 2 Report

Question #1) This study was very interesting and successfuly showed that the rate of pulmonary complication was lower in LG than in OG. This is one of the important meanings of the minimal invasiveness. I think author wrote the manuscript interestingly.

However, this study lacked the explanations for Why? Why do you think that LG group has lower pulmonary complications than OG group? Author should have expalined these.

First, the larger incisions may result worse pain. This can be the reason for the limited movement of the diaphragm and deep breathing. Compared to open TG, LATG was associated with a significant reduction in medical complications (P=0.03) with a contribution from respiratory complications (P=0.06) Pain after surgery was less serious in LATG than in open TG [reference A].

Second, Park SH [reference A] et al reported that totally laparoscopic total gastrectomy has lower rate of grade I pulmonary complications than Laparoscopy-assisted total gastrectomy. This phenomenon also can be explained as follows. The mini-laparotomy wounds of the LATG are larger and located closer to epigastrium than those of TLTG. In addition, the LATG group had a higher score of pain than the TLTG group. The larger incisions in the epigastrium and worse pain score may explain the limited movement of the diaphragm and deep breathing, followed by a decreased pulmonary function in the LATG group. All of these can explain why minimal invasiveness can make less pulmonary complications.

Therefore, please cite the two references for (1) LATG vs Open TG (to support your results) and (2) TLTG vs LATG (to suggest totally laparoscopic procedures would be less invasive than LATG) and discuss briefly.

[Reference A]. Chen K, Xu XW, Zhang RC, Pan Y, Wu D, Mou YP. Systematic review and meta-analysis of laparoscopy-assisted and open total gastrectomy for gastric cancer. World J Gastroenterol. 2013 Aug 28;19(32):5365-76.

[Reference B]. Park SH, Suh YS, Kim TH, Choi YH, Choi JH, Kong SH, Park DJ, Lee HJ, Yang HK. Postoperative morbidity and quality of life between totally laparoscopic total gastrectomy and laparoscopy-assisted total gastrectomy: a propensity-score matched analysis. BMC Cancer. 2021 Sep 11;21(1):1016.

Question #2) In methods sections, author explained that an end-to-side oesophagojejunal anastomosis was carried out with a circular stapler for total gastrectomy. Please describe the size (25 or 21mm size ?), and the name, company of the EEA (e.g. EEA 25mm, Covidien, USA).

Question #3) Regarding complications, there is no mention about the strictures or stenotic complications. If it was not recorded in your hospital, please describe these as for limitations in discussions, briefly or add it to Table 3.
